# Quantification of Lutein + Zeaxanthin Presence in Human Placenta and Correlations with Blood Levels and Maternal Dietary Intake

**DOI:** 10.3390/nu11010134

**Published:** 2019-01-10

**Authors:** Melissa Thoene, Ann Anderson-Berry, Matthew Van Ormer, Jeremy Furtado, Ghada A. Soliman, Whitney Goldner, Corrine Hanson

**Affiliations:** 1Department of Pharmacy and Nutrition, Nebraska Medicine, 981200 Nebraska Medical Center, Omaha, NE 68198, USA; 2Department of Pediatrics, University of Nebraska Medical Center, 981205 Nebraska Medical Center, Omaha, NE 68198-1205, USA; alanders@unmc.edu (A.A.-B.); matthew.vanormer@unmc.edu (M.V.O.); 3Department of Nutrition, Harvard School of Public Health, 655 Huntington Avenue, Boston, MA 02215, USA; jfurtado@hsph.harvard.edu; 4Graduate School of Public Health and Health Policy, City University of New York, 55 West 125th Street, New York, NY 10027, USA; ghada.soliman@sph.cuny.edu; 5Division of Endocrinology, University of Nebraska Medical Center, Omaha, 984130 Nebraska Medical Center, Omaha NE 68198-4130, USA; wgoldner@unmc.edu; 6College of Allied Health Professions, University of Nebraska Medical Center, 984045 Nebraska Medical Center, Omaha, NE 68198-4045, USA; ckhanson@unmc.edu

**Keywords:** lutein, zeaxanthin, carotenoid, placenta, pregnancy

## Abstract

Lutein + zeaxanthin (L + Z) are carotenoids recognized in eye health, but less is known about their status during pregnancy. While quantified in maternal and umbilical cord blood, they have never been analyzed in placenta. The purpose of this study is to quantify combined L + Z concentrations in human placenta and correlate with levels in maternal dietary intake, maternal serum, and umbilical cord blood. The proportions of combined L + Z were compared within diet, placenta, maternal serum, and umbilical cord blood among additional carotenoids (lycopene, β-cryptoxanthin, α-carotene, and β-carotene). This Institutional Review Boardapproved cross-sectional study enrolled 82 mother-infant pairs. Placenta, maternal serum, and umbilical cord blood samples were analyzed for carotenoids concentrations. Mothers completed a food frequency questionnaire and demographic/birth outcome data were collected. L + Z were present in placenta, median 0.105 micrograms/gram (mcg/g) and were significantly correlated with maternal serum (*r* = 0.57; *p* < 0.001), umbilical cord blood levels (*r* = 0.49; *p* = 0.001), but not dietary intake (*p* = 0.110). L + Z were the most prevalent in placenta (49.1%) umbilical cord blood (37.0%), but not maternal serum (18.6%) or dietary intake (19.4%). Rate of transfer was 16.0%, the highest of all carotenoids. Conclusively, L + Z were identified as the two most prevalent in placenta. Results highlight unique roles L + Z may play during pregnancy.

## 1. Introduction 

There are over 600 types of carotenoids—fat-soluble pigments that provide color to plants—found in nature [1]. Though not always deemed essential for life, they may enhance body functioning and health outcomes. The most common carotenoids found in the human diet and body tissues include α-carotene, β-carotene, lutein, zeaxanthin, lycopene, and β-cryptoxanthin [2]. Much attention has been given to the provitamin A carotenoids (β-carotene, α-carotene, and β-cryptoxanthin), which can be converted to retinol in the body and have Vitamin A activity. However, less attention has focused on the non-provitamin A carotenoids (lutein, zeaxanthin, and lycopene).

Lutein and zeaxanthin, along with β-cryptoxanthin, are more specifically known as xanthophyll carotenoids, defined as “any of several yellow to orange carotenoid pigments that are oxygen derivatives of carotenes” [3]. Of these carotenoids, isomers lutein and zeaxanthin are most well-recognized as protective agents against macular degeneration in older populations [4], due to selective uptake within the central region of the eye [5] where they filter blue light [5,6,7]. However, emerging evidence has identified associations between blood lutein and zeaxanthin levels and improved cognition in adults [8,9]. Though less is known about lutein and zeaxanthin in early life, they have been identified to consist of 59% of the carotenoids in brain tissue of children less than 1.5 years of life [10]. Further research has also identified lutein and zeaxanthin as two of the most prevalent carotenoids in human milk [11,12], which suggests neonatal-specific benefits conferred during a period of rapid growth and brain development. Preliminary evidence to support this stems from research by Cheatham et al. who reported a relationship between lutein (and choline) levels in maternal breast milk at 3–5 months postpartum and enhanced infant memory and cognitive outcomes [13]. Additional studies have found varying results during pregnancy and infancy. For example, increased maternal lutein levels at mid-gestation have been associated with lower relative risk of developing both early and late-onset preeclampsia [14]. Lutein with or without zeaxanthin supplementation has been found to have no effect on the development of retinopathy of prematurity in preterm infants born <32 weeks gestational age [15,16,17], but was found to reduce progression to severe retinopathy in infants <33 weeks gestational age [18]. With these considerations, special interest has been taken in pregnant and neonatal populations.

Lutein and zeaxanthin are unable to be synthesized in humans, so blood and tissue levels rely solely on dietary intake. However, these carotenoids are not essential so there remains no unanimous consensus recommendations for daily consumption. Lutein intakes of 6 mg/day are generally encouraged to promote eye health [19]. Recommendations to promote slower progression of age-related macular degeneration stems from one large trial, the Age-Related Eye Disease Study, which supports 10 mg/day of lutein and 2 mg/day of zeaxanthin [4]. There appears to be limited toxicology information available, but there is “strong safety evidence” with lutein doses of 20 mg/day [20]. Despite safe higher end dosing, most research in the United States report average intakes closer to 1–3 mg/day for most Americans [21]. Lutein and zeaxanthin are both available in supplement form, though are not routinely added to popular child, adult, or prenatal multivitamin formulations. Highest natural sources of lutein and zeaxanthin are found in fruits and vegetables, with primary sources as dark green vegetables. Examples of combined lutein + zeaxanthin (L + Z) content (in mg) per 100 grams of food includes 18.9 paprika spice, 12.2 spinach, 6.3 kale, 2.9 pistachio nuts, 2.5 green peas, 2.3 romaine lettuce, 2.1 summer squash, 1.6 Brussel sprouts, 0.6 yellow corn [22].

Current literature reports quantitative lutein and zeaxanthin tissue concentrations primarily within the blood, liver, kidney, adipose tissue, and brain [1]. However, levels have also been detected in multiple other areas of the body including the skin, breast, uterus, ovary, testes, adrenals, pancreas, spleen, heart, and thyroid [23,24,25]. Lutein and zeaxanthin have been identified in maternal and fetal blood, but transfer rates remain relatively low at 15.1–29.4% [26,27,28,29]. Only limited research has assessed carotenoid presence in placenta, but never of lutein or zeaxanthin [30]. Analysis of placental tissue to more fully understand carotenoid transfer remains advantageous considering research by Palan et al. who reported inconsistent carotenoid responses between placental tissue and maternal serum in preeclamptic mothers [30]. Furthermore, no study has simultaneously compared placenta, maternal, and fetal blood carotenoid levels in conjunction with maternal dietary carotenoid intake. 

The primary purpose of this study is to firstly quantify L + Z presence in human placenta. A secondary aim will seek to determine correlations between L + Z levels in placenta and levels in maternal serum, umbilical cord blood, and maternal dietary intake to more fully understand carotenoid transfer. Further aim will be to determine the relative proportions of L + Z levels in placenta, umbilical cord blood, maternal serum, and maternal dietary intake compared to other carotenoids (lycopene, β-cryptoxanthin, α-carotene, and β-carotene) to enhance knowledge of carotenoid status during pregnancy. The final aim is to identify if there are differences L + Z blood or placental levels based on birth outcomes.

## 2. Methods

### 2.1. Study Design and Participants

After Institutional Review Board approval through the University of Nebraska Medical Center, mother-infant pairs were screened and approached for informed written consent. Eligibility requirements included mothers at least 19 years of age who were free of liver, kidney, or gastrointestinal disease that influences nutrient absorption. Included mothers delivered at least one live-born infant at Nebraska Medicine hospital (Omaha, NE, USA) in 2017 who did not have congenital abnormalities or an inborn error of metabolism. No infants deemed ward of state were approached for consent to participate, per state law [31]. Mothers with multiple gestation were eligible for inclusion. Both term-born and preterm infants were included. No incentives were offered so as not to influence decision to participate.

### 2.2. Biological Samples Collection

A placenta, maternal blood, and infant umbilical cord blood sample were taken at time of delivery in enrolled mother-infant pairs to send for carotenoid analysis. Analyzed carotenoids were combined L + Z, lycopene, β-cryptoxanthin, α-carotene, and β-carotene. Placenta samples were fixed in a 10% formalin solution. Maternal blood samples were collected as part of a routine blood draw. Umbilical cord blood is routinely collected at all deliveries and stored in the hospital laboratory for use as needed. Both maternal and umbilical cord blood samples were light protected and frozen immediately once attained by the research team to preserve samples. Goal volumes for samples were 10 g placenta, 1 mL maternal blood, and 5 mL cord blood. 

### 2.3. Carotenoid Laboratory Analysis

The Nutritional Biomarker Lab at the Harvard T. H. Chan School of Public Health conducted analysis of lutein + zeaxanthin, β-cryptoxanthin, lycopene, α-carotene, and β-carotene, using reversed-phase high-performance liquid chromatography methods as described by El-Sohemy, et al. [32] with some modifications. Placenta tissue samples were weighed and then homogenized by mechanical pulverization (Polytron PT1200, Kinematica AG, Lucerne, Switzerland) in distilled, deionized water to form an aqueous slurry. The placental slurry and the plasma samples were mixed with ethanol containing rac-Tocopherol (Tocol) as an internal standard, extracted with hexane, evaporated to dryness under nitrogen, and reconstituted in ethanol, dioxane, and acetonitrile. Samples were quantitated by high-performance liquid chromatography (HPLC) on a Restek Ultra C18 150 mm × 4.6 mm column, 3 μm particle size encased in a Hitachi L-2350 column oven to prevent temperature fluctuations, and equipped with a trident guard cartridge system (Restek, Corp. Bellefonte, PA, USA). A mixture of acetonitrile, tetrahydrofuran, methanol, and a 1% ammonium acetate solution (68:22:7:3) was used as mobile phase, flow rate 1.1 mL/min, using a Hitachi Elite LaChrom HPLC system comprised of an L-2130 pump in isocratic mode, an L-2455 Diode Array Detector (monitoring at 300 nm and 445 nm), and a programmable AS-2200 auto-sampler with chilled sample tray. The system manager software (D-7000, Version 3.0) was used for peak integration and data acquisition (Hitachi, San Jose, CA, USA). The minimum detection limits (MDLs) in plasma are (μg/L) 3.86 for lutein + zeaxanthin, 3.88 for β-cryptoxanthin, 5.44 for lycopene, 4.24 for α-carotene, and 4.80 for β-carotene. Each batch of samples run included two replicates each of a two-level plasma pool sample set. Internal quality control was assessed through these four control samples. Quality control is also assessed externally by participation in the standardization program for carotenoid analysis through the National Institute of Standards and Technology, United States of America. Placenta carotenoid levels were reported in micrograms/gram (mcg/g) and blood carotenoid levels were reported in micrograms/Liter (mcg/L).

### 2.4. Dietary Intake and Birth Data Collection

The validated Harvard Willett food frequency questionnaire [33], which includes four pages questioning usual dietary intake and dietary supplement use over the past one year, was completed by mothers at time of hospitalization for delivery. Questionnaire results provided data for this study on average daily calorie, carbohydrate (g), protein (g), fat (g), and carotenoid (mg) intake. 

Additional birth outcome information was collected for mothers and infants at time of delivery from the online electronic health system. Data collected for mothers included years of age, mode of delivery (vaginal vs. Cesarean), diagnosis of any form of diabetes (Type 1, Type 2, or gestational) during pregnancy (yes/no), and diagnosis of preeclampsia (yes/no). Data collected for infants included gender, gestational age at birth (in weeks and days), preterm birth (yes/no; defined as gestational age <37 weeks at birth [34]), birth weight (g), birth head circumference (centimeters), birth length (centimeters), Apgar score at 1 and 5 min of life, respiratory distress syndrome (RDS) (yes/no), and newborn intensive care unit (NICU) admission (yes/no).

### 2.5. Statistical Analysis

All statistical analyses were conducted via SPSS software (International Business Machines Corporation, Armonk, New York, USA). As the primary aim of this study was to analyze combined L + Z concentrations in placenta, all mother-infant pairs were included if placenta analysis was available. Inclusion of all placenta data resulted in incomplete data reported for remaining continuous and categorical variables if they were not available. 

As histograms for L + Z levels (in placenta, maternal serum, umbilical cord blood, and maternal dietary intake) revealed right-skewed data that was not normally distributed, descriptive statistics for numerical values of all carotenoids (L + Z, lycopene, β-cryptoxanthin, α-carotene, and β-carotene) are reported with median, interquartile range (IQR), minimum, and maximum. To maintain consistency, all other continuous variables are reported in similar descriptive statistics. The median value for each carotenoid was divided by the sum of all carotenoid medians for a specific sample (placenta, maternal serum, umbilical cord blood, maternal dietary intake), then multiplied by 100 to obtain the proportion. Frequencies and proportions for categorical variables were calculated. 

The Mann-Whitney U test was used to compare median values for continuous variables between two groups. Spearman’s correlation coefficients were used to assess correlations between two continuous variables. Multivariate linear regression modeling with backwards elimination was used to predict how maternal dietary intake of L + Z would change maternal serum values of L + Z. A *p*-value of <0.05 was considered statistically significant for all analyses. 

## 3. Results

### 3.1. Maternal and Infant Outcomes

In 2017, there were 223 mother-infant pairs approached for study consent. Sixty-nine mothers declined participation. Of the 154 enrolled, 82 mother-infant pairs were included in this analysis as a result of placenta data available. There were no multiple births in this sample. Continuous response outcome data for mother-infant pairs are displayed in Table 1. Categorical response outcome data and frequencies are displayed in Table 2.

### 3.2. Carotenoid Concentrations

Maternal dietary intake of carotenoids is listed in Table 3. Placental, maternal serum, and umbilical cord blood carotenoid levels are displayed in Table 4. 

### 3.3. Carotenoid Proportions

Proportions of carotenoids in dietary intake, placenta, and blood levels are listed in Table 5 for comparison. Median percent of umbilical cord blood carotenoid levels compared to maternal levels are also displayed in Table 5. Non-provitamin A carotenoids (L + Z and lycopene) consisted of 58.1% of maternal dietary intake, followed by 69.0% maternal serum, 75.7% placenta, and 65.3% umbilical cord blood. Xanthophyll carotenoids (L + Z and β-cryptoxanthin) consisted of 20.4% maternal dietary intake, 30.4% maternal serum, 56.6% placenta, and 55.3% umbilical cord blood.

### 3.4. Correlations

There were no correlations between dietary intake or placental or blood levels with maternal age, infant Apgar score at 1 and 5 min of life, and infant birth anthropometric measurements (for weight, length, and head circumference). Maternal age as a continuous variable was not correlated with rate of L + Z transfer from maternal serum to umbilical cord blood.

All placental and blood L + Z levels were significantly correlated with one another. Placental levels were more strongly correlated with maternal serum (*r* = 0.57; *p* < 0.001) than umbilical cord blood (*r* = 0.49; *p* < 0.001). The strongest correlation between all levels was between maternal serum and umbilical cord blood levels (*r* = 0.65; *p* < 0.001).

Dietary intake of L + Z was not correlated with placental concentrations (*r* = 0.18; *p* = 0.110), but correlated significantly with maternal L + Z serum levels (*r* = 0.30; *p* = 0.007) and approached significance with umbilical cord blood (*r* = 0.20; *p* = 0.086). Dietary intake of L + Z was correlated with increased calorie (*r* = 0.40; *p* < 0.001), carbohydrate (*r* = 0.37; *p* = 0.001), fat (*r* = 0.42; *p* < 0.001), and protein (*r* = 0.39; *p* < 0.001) intake. However, intake of any specific macronutrient was not correlated with placental or blood L + Z levels. 

Gestational age at birth correlated with L + Z levels in placenta (*r* = 0.26; *p* = 0.019), maternal serum (*r* = 0.29; *p* = 0.010), and umbilical cord blood (*r* = 0.41; *p* < 0.001), but not maternal dietary intake (*p* = 0.160). After excluding preterm infants (*N* = 2), correlations still remained significant. Gestational age at birth also correlated with the percent of L + Z levels in umbilical cord blood compared to maternal serum (*r* = 0.31; *p* = 0.008).

### 3.5. Comparison of Medians

There was no difference in median combined L + Z levels for dietary intake, placenta, maternal serum, umbilical cord blood between categorical responses for infant gender, NICU admit, RDS, preterm birth, mode of delivery, or maternal diabetes. Notably, sample sizes were small for mothers with diabetes and infants with RDS, preterm birth, and NICU admit.

## 4. Discussion

### 4.1. Placental Levels

Combined L + Z was present in placenta tissue and able to be quantified (0.105 mcg/g). L + Z levels in placenta samples are within previous reported ranges in liver (0.06–6.9 mcg/g), kidney (0.05–5.9 mcg/g), and lung (0.05–1.3 mcg/g) [1]. However, upper end placenta ranges were quantified at 0.46 mcg/g compared to these other tissues that exceeded >1 mcg/g. Comparatively, conversions of placenta L + Z concentrations to nanograms/g (ng/g) (range 30–460) remained similar or higher than previous reports of lutein levels within the epithelial (44.1 ng/g) and nuclear eye layers (15.1 ng/g) [35]. Comparisons to retinal tissue cannot be made as concentrations are reported per m^2^. In conversion, placental concentrations would range from 56–802 picomol/g (pmol/g) (median 220 pmol/g). Placenta levels in this study are primarily higher than reported adipose tissue (268.5–456.3 pmol/g) [36] and brain concentrations (0–176.4 pmol/g lutein and 52.9 pmol/g zeaxanthin) [9,10,37]. Notably, all non-provitamin A carotenoids were detected in each placenta sample. Alternatively, a few placenta samples detected no quantitative values of β-cryptoxanthin or α-carotene, both provitamin A carotenoids.

Combined L + Z were identified as the most prevalent analyzed carotenoids in placental tissue at 49.1%, nearly double the second highest carotenoid (lycopene) at 26.6%. Increased concentrations of L + Z in placental tissue, compared to other analyzed carotenoids, may be justified by their structural properties. Firstly, lutein and zeaxanthin reportedly demonstrate higher membrane solubility compared to less polar carotenoids as a result of their end hydroxyl groups [1]. In an example from review by Widomska and Subczynski, solubility thresholds are estimated at 10 mole percent for zeaxanthin and 15 mole percent for lutein in fluid-phase model membranes [38]. Comparatively, this threshold is around 0.5 mole percent for β-carotene [38]. In addition, the hydroxyl groups on lutein and zeaxanthin’s cyclic ends allow them to be oriented perpendicular within the membrane bilayer, whereas non-polar carotenoids are more randomly oriented [38]. This orientation subsequently allows them to be affixed more firmly within the membrane layer. 

While structural properties of lutein and zeaxanthin increase potential for uptake, it remains uncertain if there is an element of selective uptake by placenta. For example, presence and orientation of L + Z within the placental membrane may improve its physical strength and stability [38,39]. Similarly, placental damage by lipid perioxidation may be decreased by L + Z scavenging free radicals both within and around the membrane [38,40]. Likewise, transmembrane orientation of L + Z increases hydrophobicity of the inner bilayer, which influences membrane permeability [38]. It is hypothesized that improved placental structure and selective permability of substrate from maternal blood is favorable in the protection and nourishment of the developing fetus. Multiple research has demonstrated that disruptions in placenta function negatively impact pregnancy outcomes, including fetal growth restriction and increased risk of stillbirth [41]. Poor functioning likely leads to multiple disruptions, but with main alterations being in nutrient transfer and oxygen exchange. Later consequences to the offspring include increased risk of developmental delay or autism and fetal programming that may result in chronic disease such a hypertension, coronary artery disease, or Type 2 diabetes [42,43,44]. If these adverse outcomes can be prevented or diminished by improved placental composition, uptake of lutein and zeaxanthin may be a mechanism of biological adaptation. Lastly, placental L + Z concentrations were not altered in instances of preterm birth or maternal diabetes, but it must be noted that sample sizes were small. 

Placental concentrations were more strongly correlated with maternal serum than umbilical cord blood levels. This finding is justified by the route of L + Z transfer from food to mother to infant, as maternal blood flows to placenta providing a source of L + Z for uptake. Uniquely, umbilical cord blood L + Z levels were more strongly correlated with maternal serum levels than placental concentrations, despite never mixing. However, while the placenta directly passes nutrients into the fetal blood circulation, the original supply of lutein and zeaxanthin sources from maternal blood. Correlations with umbilical cord blood identify how many components must function properly for the fetus to receive adequate nutrition from mother.

### 4.2. Maternal and Umbilical Cord Blood Levels

L + Z were detected in all maternal and umbilical cord blood samples. Maternal serum levels of L + Z in this study (range 25.7–605.5 mcg/L) were consistent with the range of previous reports (48–698 mcg/L), with the exception of a lesser low-end value [1]. Umbilical cord blood L + Z levels in this study (converted to median 0.06 micromol/L (mcmol/L) or 34.8 ng/mL) were lower than previous reports of 0.10 mcmol/L [26] in umbilical cord blood, but higher than levels of 24.2 ng/mL in newborn serum [28]. Results indicate a maternal to fetal transfer rate of L + Z at 16.0%, higher than any other carotenoid in this study but still on lower than most past literature reports of 15.1–29.4% [26,27,28,29]. Notably however, L + Z consisted of 18.6% of carotenoids in maternal blood, but 37.0% in umbilical cord blood. Despite higher proportions in umbilical cord blood, there were no differences in levels for the small number of infants with RDS or NICU admission.

One theory for higher proportions in umbilical cord blood is based on transportation of carotenoids in blood by lipoproteins. It has been reported that most hydrocarbon carotenoids are preferentially carried by low density lipoprotein (LDL) cholesterol, whereas polar xanthophyll carotenoids like L + Z have more of an ability to be carried by high density lipoprotein (HDL) cholesterol [1,45,46]. In example, a small study by Wang et al. (*N* = 12) reported HDL to carry approximately 52% of lutein and 44% of zeaxanthin, whereas only 20–25% of lycopene, α-carotene, and β-carotene [45]. Additionally, it must be noted that adults and neonates have different ratios of blood lipoproteins. In example, large cohort data from American adults (*N* = 424,201) show higher blood LDL levels compared to HDL, often at a ratio of 2:1 [47]. Alternatively, neonates have more circulating HDL, with summary data from Woollett and Heubi reporting ratios closer to 1:1 [48]. Consequently, a higher blood ratio of HDL compared to LDL in neonates, with bound L + Z, would explain higher presence in umbilical cord blood compared to LDL-bound carotenoids. These results ultimately highlight the unique role HDL cholesterol plays in L + Z transportation and provides rationale for why bound carotenoid proportions may vary between infant and maternal blood. This theory is further supported by results in this study in which all xanthophyll carotenoids (L + Z, β-cryptoxanthin) increased proportions from maternal to umbilical cord blood, whereas non-xanthophyll carotenoids either maintained or decreased in proportions.

### 4.3. Dietary Intake

Increasing macronutrient intake in this study was correlated with increased L + Z intake, illustrating higher carotenoid intake from increased food consumption. However, maternal dietary intake of L + Z was only correlated with maternal serum levels, but not placental or umbilical cord blood levels. This finding is unique in context that L + Z levels in maternal serum are significantly correlated with levels in placenta and umbilical cord blood. These results only contribute to our current knowledge base of nutrient transfer from food to mother to infant. While placenta and umbilical cord blood L + Z levels rely significantly on maternal blood levels, maternal serum levels remain the most sensitive to modifiable maternal lifestyle factors like dietary intake. This correlated chain of carotenoid transfer highlights how heavily dependent infants are on their mothers for receiving nutrients.

Maternal dietary intake of L + Z averaged 2.48 mg/day, which remain inadequate to meet varying recommendations that promote eye health (minimum 6–10 mg/day lutein + 2 mg/day zeaxanthin) [4,19]. In fact, only 6.3% (5/80) of mothers consumed ≥10 mg/day of combined L + Z, but 11.3% (9/80) mothers consumed <1 mg/day. While there are no unanimous recommendations for daily L + Z consumption, average maternal intake in this population of Midwestern mothers is anticipated to be low. The Centers for Disease Control and Prevention reported in 2015 only 11.4% of Nebraskans (*N* = 13,771) consumed recommend fruit intake with only 7.9% for recommended vegetable intake [49]. Additionally, it remains uncertain if maternal dietary intake was not high enough in this sample to significantly influence placenta or umbilical cord blood levels, as L + Z levels fractionate as maternal blood distributes them to various body tissues.

### 4.4. Strengths and Limitations

The primary strength of this study is that placenta carotenoid levels have never been quantified for L + Z presence. Furthermore, no study has assessed placental levels concurrent with maternal dietary intake, maternal serum, and umbilical cord blood levels to observe full maternal-fetal transfer dynamics. Furthermore, jointly comparing the most six prevalent carotenoids within the American diet allows comparisons of proportions and transfer rates. 

Limitations of this study include that usual dietary intake is self-reported, despite using a validated food frequency questionnaire. It also remains difficult to compare previous studies as lutein and zeaxanthin may be analyzed separately or conjointly. Additional factors were not reported in this study that may influence carotenoid levels, such as smoking or maternal body mass index size.

## 5. Conclusions

Carotenoids L + Z were quantified in human placenta at a median of 0.105 mcg/g and were significantly correlated with combined L + Z levels in maternal serum and umbilical cord blood, but not maternal dietary intake. Only maternal serum levels were significantly correlated with maternal dietary intake. L + Z were also the most prevalent carotenoids in placenta and umbilical cord blood, when compared alongside α-carotene, β-carotene, lycopene, and β-cryptoxanthin. L + Z rate of transfer from maternal to fetal blood was 16.0%, the highest of all carotenoids. Conclusively, L + Z consisted of 49.1% of placental carotenoids, highlighting the unique roles L + Z may play during pregnancy. Future research is needed to identify specific benefits conferred to the developing infant. 

## Figures and Tables

**Table 1 nutrients-11-00134-t001:** Continuous responses of maternal and infant outcomes.

	*N*	Median	Interquartile Range	Minimum	Maximum
Gestational Age at Birth ^a^	82	39 + 5	1 + 4	32 + 1	42 + 1
Birth Weight (g)	82	3512	535	1580	4617
Birth HC (cm)	81	34.9	1.3	24.5	38.1
Birth Length (cm)	81	50.8	3.8	34.3	55.2
Apgar Score 1 min	80	8.0	0.3	1.0	9.0
Apgar Score 5 min	80	9.0	0.0	5.0	9.0
Maternal Age (years)	81	29	9	19	43
Maternal Calorie/day Intake	80	2052	968	712	5758
Maternal Protein/day Intake (g)	80	79	37	22	205
Maternal Fat/day Intake (g)	80	78	41	16	228
Maternal Carbohydrate/day Intake (g)	80	261	139	114	772

^a^ Reported in “weeks + days”.

**Table 2 nutrients-11-00134-t002:** Categorical responses of maternal and infant outcomes.

	*N*	Frequency (%)
Infant Gender	82	30 female (36.6%) 52 male (63.4%)
NICU Admit	82	7 (8.5%)
RDS	82	3 (3.7%)
Preterm Birth	82	2 (2.4%)
Mode of Delivery	81	66 vaginal (81.5%) 15 Cesarean (18.5%)
Maternal Diabetes	81	6 (7.4%)
Preeclampsia	81	0%

NICU: newborn intensive care unit.

**Table 3 nutrients-11-00134-t003:** Maternal dietary intake of carotenoids (mg/day).

*N* = 80	Median	Interquartile Range	Minimum	Maximum
L + Z	2.48	2.25	0.27	17.78
Lycopene	4.94	3.69	1.27	21.29
β-cryptoxanthin	0.12	0.16	0.02	0.62
α-carotene	0.49	0.56	0.01	4.61
β-Carotene	4.72	3.34	0.94	27.45

**Table 4 nutrients-11-00134-t004:** Carotenoid levels in placental tissue (mcg/g) and blood samples (mcg/L).

*N* = 82	Median	Interquartile Range	Minimum	Maximum
**L + Z**				
Placenta	0.105	0.082	0.03	0.46
Maternal Serum	220.8	122.3	25.7	605.5
Umbilical Cord Blood	34.8	29.5	14.6	144.7
**Lycopene**				
Placenta	0.057	0.05	0.01	0.16
Maternal Serum	599.1	349.4	25.8	1109.4
Umbilical Cord Blood	26.6	20.1	6.5	454.9
**β-cryptoxanthin**				
Placenta	0.016	0.01	0	0.09
Maternal Serum	140.7	138.4	20.9	530.1
Umbilical Cord Blood	17.2	15.7	4.6	683.4
**α-carotene**				
Placenta	0.008	0.01	0	0.08
Maternal Serum	42.6	55.4	2.4	1022.30
Umbilical Cord Blood	3.5	7.2	0	939.4
**β-Carotene**				
Placenta	0.028	0.04	0.002	0.23
Maternal Serum	186	287.4	7.1	3003.1
Umbilical Cord Blood	11.9	17.8	0	286.6

**Table 5 nutrients-11-00134-t005:** Proportions of median carotenoid levels within maternal diet, placenta, maternal serum, and umbilical cord blood.

	Maternal Dietary Intake	Placenta	Maternal Serum	Umbilical Cord Blood	Percent Cord: Maternal Serum
L + Z	19.4%	49.1%	18.6%	37.0%	16.0%
Lycopene	38.7%	26.6%	50.4%	28.3%	5.0%
β-cryptoxanthin	1.0%	7.5%	11.8%	18.3%	12.0%
α-carotene	3.9%	3.7%	3.6%	3.7%	6.0%
β-Carotene	37.0%	13.1%	15.6%	12.7%	5.4%

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
