# Peer review of "Quantification of Lutein + Zeaxanthin Presence in Human Placenta and Correlations with Blood Levels and Maternal Dietary Intake"

_nutrients, 2019, doi:10.3390/nu11010134_

Reviewer 1 Report

This is well-written, however, there are some suggestions including the discussing of possible health implications of these results, and the dietary intake during pregnancy and maternal serum concentrations are not discussed in any great detail.

Also, after careful analysis of the manuscript, I have however a few general comments:

-The type of formatting the information about the number of participants or correlations, and p-values  (n; r) should be changed for italics and the adding space before and after mathematic signs are necessary.

- Is reporting the percentages with one decimal places are necessary, due to the study group less than hundreds of participants?

- Please, add more about the role of carotenoids in pregnancy and neonatal period, there are much more studies investigating it than you mentioned.

- The material and methods section has to be improved:

o I suggest you add more information to a selection of participants. How many women were invited to the study? How many mothers agreed to the study? Maybe it's better to present it in the diagram - (Flowchart of sample collection), including the inclusion/exclusion criteria for the study.

o  I recommended dividing this section for more subsection: study design, participants, data collections, dietary assessment, biological sample collection, carotenoid analysis.

o  The part about biological sample has to be described more detailed.

o  The information about carotenoid analysis have to be added, including the samples preparation, and the information about HPLC analysis (used devices, reagents, standards, analysis parameters).

o  There is a lack of information about used food-composition tables, especially the data used to the carotenoid intake analysis. It was USDA data?

- There is a lack in a discussion of other carotenoids concentrations.

Line 20-21: The literature provides information about L+Z status during pregnancy and its health importance, for example, possible reducing the risk of preeclampsia.

Line 60: Only L+Z are not synthesized in humans? Please correct this information.

Line 61: Why is no consensus in the dietary recommendation? Are carotenoids are considered as essential nutrients?

Line 66: You should check whether reference 15 should be cited?

Line 69-70: What with the other than vegetables&fruits food sources of L+Z? Are they common? Add this information about others food sources.

Line 72: How much of paprika spice is usually used…? Please change this position with some other products. Food source may be presented in a table.

Line 91: Which demographic outcomes? In methods is only information about collecting data about maternal age. In the results, you reported results only for gestational age and neonatal health outcomes?

Line 94: Please, add the number of approval.

Line 134-135: There is information about breastmilk samples, but it was not mentioned earlier in the information about biological samples collection. As breastmilk samples were not analyzed I recommended deleting this part.

Line 151-158: You reported information about health, birth parameters, and maternal dietary intake, there is almost no information about a demographic characteristic of a study group, despite the maternal age and infant gender.  Please change the word ‘demographic’. Maybe it would be better to report the information about maternal dietary intake in the 3, not in table 1.

Line 156-157: There is no information about units of maternal age.

Line 174 & 194: Please, change the names of subsections. What are you investigated?

Line 174 – 193: It would be better to report the results of Spearman’s correlation analysis in the table (one for all checked variables and dietary, maternal serum, cord blood, placenta L&Z). Did you calculate the coefficients for maternal age, if no please add this?

Line 194 – 197: You have a very little number of preterm infants and RDS, which have a high influence of the results on U M-W test… Please  be careful in the interpretation of this results and rewrite this.

Line 199 and 250: Thought about changes the places of this section, and write first about  the maternal serum & cord blood, due to that you write in placental levels about correlations with maternal & cord blood levels.

Line 232 – 240: Please add more about pregnancy complication and placenta.

Line 249: You wrote this in line 238-240

Line 260: What about simple size…?

Line 290: Be careful comparing the carotenoids with essential nutrients.

Line 293-294: This should be mentioned in the results.

297 – 298: How looks the vegetable & fruit intake in your study group? Please add this results, especially that you used the FFQ method.

Line 291-301: What about other studies analyzing carotenoid intake during pregnancy?

Line 308 – 309: Please add something about the limitations of the FFQ methods. The lack of dietary recommendation is not a lack of this study. Please add information about the size of study group. Also, you didn’t have the data about maternal prepregnancy BMI, please write something about that, due to the possible influence of adipose tissue on the carotenoid status.

Author Response

Thank you for reviewing our manuscript and providing feedback.  Please see responses to all comments and suggestions.

The type of formatting the information about the number of participants or correlations, and p-values  (n; r) should be changed for italics and the adding space before and after mathematic signs are necessary.

All reported N, r, and p-values have been italicized.

Is reporting the percentages with one decimal places are necessary, due to the study group less than hundreds of participants?

Thank you for this suggestion.  Our intention was to report a detailed and accurate description of the proportions of L+Z levels within diet, blood, and placenta.  To maintain consistency in reported values, proportions for remaining outcomes (i.e. demographics and clinical outcomes) have been reported with one decimal place.

Please, add more about the role of carotenoids in pregnancy and neonatal period, there are much more studies investigating it than you mentioned.

Additional studies during pregnancy and infancy have been included in the background.

I suggest you add more information to a selection of participants. How many women were invited to the study? How many mothers agreed to the study? Maybe it's better to present it in the diagram - (Flowchart of sample collection), including the inclusion/exclusion criteria for the study.

We have added information to the Results section that reports the number of mother-infant pairs approached for consent, the number who declined, and the number included in our study.

I recommended dividing this section for more subsection: study design, participants, data collections, dietary assessment, biological sample collection, carotenoid analysis.

We have added more subsections.

The part about biological sample has to be described more detailed. The information about carotenoid analysis have to be added, including the samples preparation, and the information about HPLC analysis (used devices, reagents, standards, analysis parameters).

We have added to this section.

There is a lack of information about used food-composition tables, especially the data used to the carotenoid intake analysis. It was USDA data?

Maternal carotenoid intake was identified from a validated food frequency questionnaire, as reported in the Methods section.

There is a lack in a discussion of other carotenoids concentrations.

The focus of this manuscript is combined lutein + zeaxanthin concentrations.  Therefore, the remaining concentrations were just primarily used to calculate proportions of all carotenoids within maternal diet, placenta, and blood.

Line 20-21: The literature provides information about L+Z status during pregnancy and its health importance, for example, possible reducing the risk of preeclampsia.

We have included references in the Background that discuss lutein levels mid-pregnancy and risk of preeclampsia.

Line 60: Only L+Z are not synthesized in humans? Please correct this information.

Correct, L+Z cannot be synthesized by humans.   This was added in to clarify in the background.  Again, the focus of this manuscript is L+Z, so the remaining carotenoids are not discussed in high detail.

Line 61: Why is no consensus in the dietary recommendation? Are carotenoids are considered as essential nutrients?

As reported in the background, L+Z are not deemed essential for life and no dietary recommendations for daily intake exist for L+Z.

Line 66: You should check whether reference 15 should be cited?

After rechecking, we confirm this reference is correct.

Line 69-70: What with the other than vegetables&fruits food sources of L+Z? Are they common? Add this information about others food sources.

As there are many fruits/vegetables that contain L+Z, we cannot list them all and have therefore chosen to include primary sources as examples.  However, we have indicated that dark green vegetables are a primary source and the examples listed demonstrate this.

Line 72: How much of paprika spice is usually used…? Please change this position with some other products. Food source may be presented in a table.

Paprika has been chosen as an example due to its high natural content of L+Z.  It is listed per 100 grams.

Line 91: Which demographic outcomes? In methods is only information about collecting data about maternal age. In the results, you reported results only for gestational age and neonatal health outcomes?

“Demographic” has been removed.  The birth and clinical outcomes for both mother and infant are reported in the final paragraph in section 2.1 in the Methods and the results of all are reported in the Results section.

Line 94: Please, add the number of approval.

We have included the number of mother-infant pairs under the result section.

Line 134-135: There is information about breastmilk samples, but it was not mentioned earlier in the information about biological samples collection. As breastmilk samples were not analyzed I recommended deleting this part.

This portion was removed.

Line 151-158: You reported information about health, birth parameters, and maternal dietary intake, there is almost no information about a demographic characteristic of a study group, despite the maternal age and infant gender.  Please change the word ‘demographic’. Maybe it would be better to report the information about maternal dietary intake in the 3, not in table 1.

“Demographic” has been removed.

Line 156-157: There is no information about units of maternal age.

Units of measurement were added to the table.

Line 174 & 194: Please, change the names of subsections. What are you investigated?

Thank you for this suggestion, but the subsection titles have been maintained as they match what results are being reported in that section.

Line 174 – 193: It would be better to report the results of Spearman’s correlation analysis in the table (one for all checked variables and dietary, maternal serum, cord blood, placenta L&Z). Did you calculate the coefficients for maternal age, if no please add this?

Subsection 3.4 indicates no correlation between maternal age (as a continuous variable), and L+Z levels in diet, placenta, or blood. 

Line 194 – 197: You have a very little number of preterm infants and RDS, which have a high influence of the results on U M-W test… Please  be careful in the interpretation of this results and rewrite this.

You are correct.  We have added to this section.

Line 199 and 250: Thought about changes the places of this section, and write first about  the maternal serum & cord blood, due to that you write in placental levels about correlations with maternal & cord blood levels.

We prefer to maintain the current order with placental results discussed first.  Reasoning for this is because our primary specific aim is to quantify L+Z levels in human placental tissue.

Line 232 – 240: Please add more about pregnancy complication and placenta.

We have added to this section.

Line 249: You wrote this in line 238-240

This portion has been removed.

Line 260: What about simple size…?

We have added to this section to note the small sample sizes.

Line 290: Be careful comparing the carotenoids with essential nutrients.

Correct, thank you.

Line 293-294: This should be mentioned in the results.

These results are listed in Table 3.

297 – 298: How looks the vegetable & fruit intake in your study group? Please add this results, especially that you used the FFQ method.

We do not feel this would add significantly to the results of this study.  While fruit and vegetables are high natural sources of L+Z, we were more interested in the combined (mg/day) carotenoid intake.

Line 291-301: What about other studies analyzing carotenoid intake during pregnancy?

As indicated in the background, most Americans do not consume high amounts of L+Z in general.  Similarly as indicated in the background, this study is unique that it compares maternal dietary intake alongside blood and placental levels.

Line 308 – 309: Please add something about the limitations of the FFQ methods. The lack of dietary recommendation is not a lack of this study. Please add information about the size of study group. Also, you didn’t have the data about maternal prepregnancy BMI, please write something about that, due to the possible influence of adipose tissue on the carotenoid status.

The FFQ is a validated tool, so primary limitations from this method is self-reported intake by the mothers.  We have removed the statement about the lack of dietary recommendations.  We respectfully disagree about the sample size, as 82 mother-infant pairs is a sufficient sample size to report novel data (also N>30).  We have added a statement about BMI not being reported.

Reviewer 2 Report

I have reviewed the manuscript "Quantification of Lutein + Zeaxanthin Presence in Human Placenta and Correlations with Blood Levels and Maternal Dietary Intake" and found it to be an interesting and well-written contribution to the nutrient literature.  The background is very well done, the methods are sound and appropriate, and the conclusions are well-defended and appropriate from the results.

But for the benefit of the English or American-English reader, I have several suggestions:

Overall: suggest defining "L+Z" early in the manuscript; avoids reader fatigue with reading "lutein + zeaxanthin" hundreds of times.

Overall: suggest defining L+Z as "combined L+Z" so it's clear to the reader that the analysis is of the combination, not of separate L or Z carotenoids.

Line 21: suggest adding "Their" to the beginning of the sentence.

Line 23: replace "with levels in..." with "with levels of..."

Line 24: suggest replacing "Proportions..." with "The proportions of combined L+Z"

Line 26: suggest adding "This" before the word "IRB"

Lines 39-40: suggesting changing to: "...of carotenoids--fat-soluble pigments that provide color to plants--found in nature."

Line 50: suggest removing comma after "...eye[5]"

Line 54: suggest replacing "top..." with "most prevalent..."

Line 90: for readability, I suggest adding "The..." before "final aim..."

Table 2: suggest replacing "Maternal Diabetes" with "Gestational Diabetes" (that is the standard term)

Line 177: REQUIRED - change "were" to "was"

Lines 200 and 213: Please ensure that the reader is reminded that the L + Z is combined L+Z.

Line 213: suggest changing to "...highest carotenoid (lycopene) at 26.6%."

Line 249: This sentence lacks proper punctuation (i.e., no period at the end) and is out of place.  Perhaps remove the new paragraph and simply add to end of previous paragraph (end of line 248).

Line 252: suggest changing "...were within range of..." to "were consistent with the range of..."

Lines 308-311: suggest removing from "Ultimately..." (line 308) to "...more subjective." (line 311)  I do not think this limitation makes any sense and only distracts from the manuscript.

Line 315: suggest changing to "...correlated with combined L +Z levels..."

Again, this is a very interesting manuscript.

Author Response

Thank you for reviewing our manuscript and providing thoughtful feedback.  All suggested revisions have been implemented into the revised version with few exceptions as follows:

We have maintained the wording “Maternal Diabetes” in Table 2, but have instead included clarification within the methods section that maternal diabetes included all mothers with any form of diabetes (Type 1, Type 2, or gestational).

Line 249 has been removed as this was included in error.

Round  2

Reviewer 1 Report

I have no further comments.
Authors fulfilled most of my objections.